# A Fusion Decision-Making Architecture for COVID-19 Crisis Analysis and Management

Kuang-Hua Hu [1], Chengjie Dong [1,2], Fu-Hsiang Chen [3,*], Sin-Jin Lin [3] and Ming-Chin Hung [4]

1   Finance and Accounting Research Center, School of Accounting, Nanfang College, Guangzhou 510970, China; khhu0622@gmail.com (K.-H.H.); dongchengjie@163.com (C.D.)
2   Faculty of Management Science, Lampang Rajabhat University, Lampang 52100, Thailand
3   Department of Accounting, Chinese Culture University, Taipei 11114, Taiwan; annman1204@gmail.com
4   Department of Financial Engineering and Actuarial Mathematics, Soochow University, Taipei 10048, Taiwan; nhungg@scu.edu.tw
*   Correspondence: chenfuhsiang1@gmail.com; Tel.: +886-286-10511 (ext. 35525)

**Abstract:** The COVID-19 outbreak has had considerably harsh impacts on the global economy, such as shutting down and paralyzing industrial production capacity and increasing the unemployment rate. For enterprises, relying on past experiences and strategies to respond to such an unforeseen financial crisis is not appropriate or sufficient. Thus, there is an urgent requirement to reexamine and revise an enterprise's inherent crisis management architecture so as to help it recover sooner after having encountered extremely negative economic effects. To fulfill this need, the present paper introduces a fusion architecture that integrates artificial intelligence and multiple criteria decision making to exploit essential risk factors and identify the intertwined relations between dimensions/criteria for managers to prioritize improvement plans and deploy resources to key areas without any waste. The result indicated the accurate improvement priorities, which ran in the order of financial sustainability (A), customer and stakeholders (B), enablers' learning and growth (D), and internal business process (C) based on the measurement of the impact. The method herein will help to effectively and efficiently support crisis management for an organization confronting COVID-19. Among all the criteria, maintaining fixed reserves was the most successful factor regarding crisis management.

**Keywords:** artificial intelligence; multiple criteria decision making; decision making; crisis management; COVID-19





## 1. Introduction

In December 2019, an emerging coronavirus (SARS-CoV-2) originated from Wuhan, China, that has since become known to be a highly infectious and acute respiratory virus that poses a considerable impact and threat to humans [1,2]. Because this disease spread with extreme velocity and there was no available vaccine for it, most countries responded to this circumstance by closing public and private areas, such as schools and institutions, and announcing travel restrictions and lockdown/shutdown policies [3]. The World Health Organization (WHO) considered the situation caused by this disease to be life-threatening and declared the outbreak of SARS-CoV-2 a pandemic [4].

Lockdown or shutdown policies can help to stem the rapid human-to-human disease transmission, but they also cause supply chains to become at a near standstill, as well as break down the functioning of the financial market in its ability to channel resources to suitable places [5]. The disease sharply lowered the amount of global investment, fragmented international trade and supply chain interactions, and eroded human capital with higher unemployment rates. The United Nations Conference on Trade and Development (UNCTD) (2020) stated that the forecasted value of global merchandise trade would fall by 5.6% in 2020 versus that in 2019 [6]. The Organization for Economic Co-operation and

Development (OECD) also indicated that global gross domestic product (GDP) growth would dwindle to 2.4% from 2.9% in 2020 and have a higher chance to fall to zero in the worst-case situation [7].

Supply chain management (SCM) aims at planning, steering, monitoring, and controlling a firm's inherent operations and coordinating related business activities with supply chain (SC) partners (Council of the Supply Chain Management Professionals (CSCMP) [8]). It is widely viewed as one main economic backbone and an integral part of sustainable development for most countries. Due to the COVID-19 pandemic, business closures and lockdowns have had inevitable and adverse impacts on SCs, almost paralyzed several industries, and caused huge disturbances in supply and demand at the global and local scales. Furthermore, the survey done by Fortune (2020) pointed out that more than 94% of the top 1000 enterprises were negatively affected by this outbreak [9]. Thus, calling for more empirical studies on corporate crisis management to elucidate how enterprises confronted COVID-19 threats might establish numerous avenues to alleviate the side effects caused by this pandemic [1]. van Heok et al. [10] also indicated that corporate managers are unable to operationalize the concepts of crisis management and thus urged academia to conduct an empirical study to examine how managers are coping with COVID-19 challenges. In order to ensure the sustainable development of corporates in these uncertain times, there is an urgent requirement to exploit and realize the impact of COVID-19 on business operations so as to reform their crisis management and assessment architecture [7].

The performance of a corporate's crisis management is highly relevant to its risk-absorbing ability, which depends on how many tangible and intangible resources it owns and how it deploys the resources to suitable places to generate profit in a risky environment [11]. Corporate financial performance measures, such as returns on assets (ROA) or returns on equity (ROE), are widely adopted to represent operating performance due to the nature of these assessment measures being easy to understand, intuitive, and comprehensive. However, in today's knowledge-intensive economy, a corporate's competitive edge has shifted from financial assessment measures toward non-financial assessment measures, such as multi-skilled employees, customer loyalty, corporate culture, and managers with superior managerial ability [12]. Amado et al. [13] also suggested that moving away from unique, all-embracing measures (like financial indicators) toward several complementary measures (such as non-financial indicators) can be advantageous for performance assessment and improvement. By adopting an overarching and complementary mechanism, the multi-dimensional nature of performance and the obligation to answer to the interest of stakeholders are better emphasized. Balanced scorecards (BSCs), introduced by Kaplan and Norton (1992), are undoubtedly one of the best well-known and widely adopted frameworks for performance assessment. They can translate an organization's strategic aims and goals into a set of practical performance measures distributed among four essential categories: financial, customer, internal process, and learning and growth.

Although BSCs pose many strengths and are widely used, they are criticized as being deficient at capturing the interdependent causalities between the drivers and the outcome [14]. Decision-making and trial laboratory (DEMATEL) can be employed to gain more managerial insights and clarify the causalities between indicators and outcomes in a vague and intricate situation [15–18]. Through the joint utilization of BSCs and DEMATEL, we extended the original BSCs that merely considers unidirectional relations to interconnected relations among indicators so as to gain deeper insights for advanced management.

For an unknown domain (such as the COVID-19 outbreak), decision makers tend to gather messages as much as possible to realize the inherent reality of the domain to be analyzed. Unfortunately, too many messages confronted by users will bias their judgments and lead to improper decision outcomes. To combat this, the dominance-based rough set theory (DRST), which is an essential feature identification algorithm, can be considered. The merits of DRST are summarized as follows: (1) it explores essential indicators in a large dataset, (2) it takes ordinal properties of data related to preferences into account to reach

an outcome with a greater consensus, and (3) it represents the decision logic in a human-readable format (if–then style) and provides a platform for users to judge and examine the inherent decision rules so as to increase its acceptance by end-users. By filtering out redundant features, the users can place greater emphasis on real essential parts (i.e., red flags) without any hysteresis.

The aim of this study can be stated as follows [19,20]: (1) It developed an emerging crisis management and assessment architecture grounded in BSCs for corporate operations in response to COVID-19 impacts. (2) It equipped the BSCs with intertwining relations among indicators via DEMATEL to gain more valuable insights. (3) It identified essential risk elements associated with the key data source from expert knowledge and an in-depth literature review using DRST. As experts' judgments are subject to uncertainty and impreciseness with non-probabilistic characteristics (i.e., experts have dissimilar backgrounds and educational levels) [21], fuzzy set theory (FST) with the advantages of handling uncertainties and vagueness was utilized [22].

This paper is organized into five main sections. The introduction presents the impact of COVID-19 on corporate operations and how enterprises analyze and manage risks to respond to this harsh disruption. The second section is based on a literature review of the relevant studies. The third section outlines the applied methodologies. The fourth section depicts the experimental results. The fifth section discusses the managerial implications. The research work then concludes with suggested future work directions and advancements.

## 2. Research Background

The crises that enterprises face, including those caused by biochemical technology and human beings, energy crises caused by wars, or those due to natural disasters and infectious diseases, can seriously damage or ruin their goal of sustainable development. In order to appropriately react to the impacts of COVID-19, this study relied on BSCs' four perspectives—"financial", "customer", "internal business process", and "learning and growth"—and established a COVID-19 crisis management evaluation architecture to identify the key risk factors. By following this framework, decision makers can prioritize the risk factors based on their essence and target to handle them appropriately so as to recover quickly from unforeseen market disturbances.

### 2.1. Crisis Management

Crisis management can be viewed as the key to the existence of an enterprise that is normally required to be made under time pressure to avoid threatening the enterprise's value. Because potential threats can destroy an enterprise's profitability and growth, management is required to recognize and identify such possible threats in advance and conjecture suitable avenues to properly react to market and/or business disturbances. If no action is taken, then a sudden encounter with a crisis will make the situation worse and irreversible [23–26]. Crisis management, which is a systematic method and a continuous process, requires management to constantly make modifications and corrections due to changes in core technologies, environments, and stakeholders of enterprises—that is, the concept of crisis management needs to be updated and renewed to keep up with the current development of modern business management. Doing so can help the whole enterprise avoid any crisis bursting out and allow for effectively managing and responding to an actual risk outbreak [27,28].

Crises can bring significant detrimental damage and losses to corporate operations, and nearly all outcomes appear in financial reports. Thus, numerous researchers have concentrated heavily on constructing financial pre-warning models via analyzing financial reports in order to gain more valuable insights [29–33]. Such systems can analyze and forecast financial conditions as well as the production and operating activities of enterprises according to the signals sent by abnormal indicators. They also help to identify potential risks in enterprises' production, operations, and management activities and can

alert corporate operators and managers before financial crises arise so that they can take preventive measures in advance. Financial analysis systems identify signs and symptoms of corporate financial deterioration within a very short period and remind operators to make appropriate preparations. They help management look for the causes of financial deterioration, facilitate their ability to take effective measures to control the spread of a financial crisis and reduce losses, and provide a safe and secure financing environment for corporate development [34]. Based on the aforementioned studies, crises can be roughly divided into two different types: one is relevant to an enterprise's internal processes or operation policies, while the other is relevant to the external environment. No matter what type of crisis it is, managers need to adopt effective management tools to cope with it so as to avoid unnecessary bad consequences.

### 2.2. Balanced Scorecards (BSCs)

The concept of balanced scorecards (BSCs) was first proposed by KPMG in order to design a performance appraisal system for Apple Inc. and was later developed by Professor Robert Kaplan at Harvard University and David Norton in business circles. After summarizing the successful experience of 12 companies in developing performance management systems, Kaplan and Norton proposed and promoted BSCs to the world. After that, they published articles on BSCs in the Harvard Business Review in succession. *Balanced Scored Card-Measures that Drive Performance* [12] first pointed out the benefits gained by companies in using BSCs for performance appraisal. *Putting the Balanced Scorecard to Work* [35] then stated that the basis of performance appraisal indicator selection is the key success factor of corporate strategies. *Using the Balanced Scored Card as a Strategic Management System* [14] next solved two problems: one is the importance of BSCs, as the book discusses in detail their importance as a strategic management tool for corporate strategic practice; another is the framework, as the book outlines and explains the framework of BSCs as a strategy and performance management tool.

BSCs are a tool for measuring company performance in four dimensions: finance, customers, employees, and internal processes of an enterprise, and a strategy map can build a framework for the strategic goals of the four dimensions of an enterprise [36]. The strategy map is an extension of the BSCs, which shows how the company transforms the company's different assets into the company's desired results, and the company can develop its strategy map according to its different goals. With the four dimensions of the BSCs, a strategy map model is created for various industries and acts as a reference for each enterprise to implement strategies, which can not only focus on the company's strategy but also significantly enhance the cooperation and coordination within the company [35,36]. Beasley et al. [37] stated that the validity and effectiveness of BSCs can be strengthened by incorporating them into enterprises' crisis management. By doing this, we can link crisis management to enterprises' strategic performance evaluation, as well as assist managers in targeting profit maximization under anticipated risk exposure and expanding the scope of crisis management.

### 2.3. Balanced Scorecards and COVID-19 Crisis Management

Because BSCs can integrate internal and external risk factors [38], this study aimed at developing an effective crisis management architecture that relies upon BSCs to assist enterprises to supervise and carry out crisis management strategies so as to prevent avoidable losses, as well as yield a suitable direction for managers to plan for any follow-up remedies. Chipriyanov and Chipriyanova [39] indicated that enterprises' crisis management strategies place top priority on sustainable development and operations. However, the foundation and main source for the survival and stability of enterprises is how many resources they own or gain that can be converted into financial resources. Thus, enterprises need to adopt defensive accounting policies to increase their risk-absorbing ability. Mobasher [40] noted that suitable crisis management can be inferred through systematic observation and crisis risk analysis. In this study, the essential indicators for crisis management were decided via a

literature review. TOPSIS was then executed to prioritize the selected crisis indicators based on their essence and then inserted into BSCs to form a crisis management architecture.

The spread of COVID-19 has caused a severe global lockdown, and enterprises of all sizes have been affected by it to various degrees. The COVID-19 outbreak has led to a decline in orders and a sharp drop in revenues for many firms. Under financial pressure from cash flows and payments, such as wages, rents, and interests, large enterprises may suffer losses spanning months or even quarters but can recover after pandemic mitigation and then try to make up for those losses. However, many small- and medium-sized enterprises (SMEs) may not be so lucky, as a wave of bankruptcies may be triggered if they have financial problems and run into uncontrolled losses. In order to help enterprises properly handle the impacts of COVID-19, this study expanded the existing crisis management study via the BSC architecture, further explored COVID-19 crisis management, reviewed existing literature, and concluded with practical ideas. The indicators for a COVID-19 crisis management architecture can be categorized into four dimensions: "financial sustainability", "customer and stakeholders", "internal business process", and "enablers' learning and growth" [41–46]. The dimensions and assessing criteria are presented in Table 1.

**Table 1.** Dimension and criteria of COVID-19 crisis management for the pre-test questionnaire.

| Dimension (A): Financial Sustainability | |
|---|---|
| **Criteria** | |
| $a_1$: Controlling fixed costs | $a_2$: Developing as many financial resources as possible |
| $a_3$: Maintaining a good relationship with banks | $a_4$: Maintaining fixed reserves |
| $a_5$: Increasing asset efficiency | $a_6$: Improving return on equity |
| $a_7$: Increasing revenues | $a_8$: Business growth |
| $a_9$: Actively applying for grants | |

| Dimension (B): Customer and Stakeholders | |
|---|---|
| **Criteria** | |
| $b_1$: Increasing the delivery speed | $b_2$: Enhancing product functions |
| $b_3$: Strengthening advertising promotion | $b_4$: Managing customer relationships |
| $b_5$: Promoting product updates | $b_6$: Promoting mold design |
| $b_7$: Improving product quality and durability | $b_8$: Serving customers using dedicated customer service teams |
| $b_9$: Analyzing reasons for returns, exchanges, and customer complaints | |

| Dimension (C): Internal Business Process | |
|---|---|
| **Criteria** | |
| $c_1$: Strengthening inventory management | $c_2$: Deploying emergency decision-making teams |
| $c_3$: Completely importing activity value management | $c_4$: Raising product standards |
| $c_5$: Improving manufacturing process and delivery time | $c_6$: Making full use of machines |
| $c_7$: Enhancing new product research and development | $c_8$: Systematically recording manpower and hours of services during crises |

| Dimension (D): Enablers' Learning and Growth | |
|---|---|
| **Criteria** | |
| $d_1$: Motivating employees to have diversified skills | $d_2$: Investing in educational training |
| $d_3$: Designing complete performance reward systems | $d_4$: Improving employee efficiency |
| $d_5$: Strengthening human cost control | $d_6$: Strengthening brand image training |
| $d_7$: Improving employee satisfaction | $d_8$: Strengthening corporate culture |

*2.4. Intelligent Models in Crisis Management*

A statistical-based model has been widely adopted to handle crisis management tasks with satisfactory performance [47], but it also comes with a critical challenge, as it needs to obey strict statistical assumptions that are difficult to satisfy in real-life applications. With the great advancement in information technology and the Internet, artificial intelligence (AI)-based models can overcome the aforementioned tasks and demonstrate their usefulness and effectiveness in many domains. Santoso and Wibowo [48] developed a financial risk pre-warning model via a support vector machine (SVM) (one type of AI-based model), demonstrating that its forecasting quality is better than statistical-based models.

Essential feature identification is another fruitful research domain for AI applications. The data for crisis management are largely collected from financial reports or documents. However, most financial messages are contaminated by some degree of errors such as selective accounting principles or the adoption of dissimilar estimation methods that can bias the user's decision making. To combat this, an essential feature identification approach called the dominance-based rough set theory (DRST), which identifies the most valuable feature subset without deteriorating the model's forecasting performance and eliminates the data storage requirement, was adopted.

## 3. Methodologies

This study developed a fusion decision architecture (called fuzzy multiple rule-based decision-making architecture: FMRDM) that integrates DRST and Fuzzy-DEMATEL (FDEMATEL) for COVID-19 crisis management to target the goal of sustainable development. Based on an in-depth literature review, we identified the risk factors for COVID-19 crisis management. To filter out irrelevant and redundant risk factors, we employed DRST. Sequentially, the selected risk factors were fed into FDEMATEL to extract the intertwining and interactive relationships between factors (DEMATEL), which is equipped with FST to assist decision makers to convert the inherent vagueness and hesitation of human thinking into crisp numbers (Xu et al., 2020). By adopting FMRDM, we can depict the network structure of factors/criteria, prioritize the essential factors/criteria for modification/revision, allocate appropriate resources to suitable locations, strengthen enterprises' COVID-19 crisis management abilities, and solidify their operations so as to overcome huge business disturbances. FMRDM is illustrated in detail as follows.

**Stage 1: DRST is used to screen out the irrelevant criteria and form a finalized questionnaire.**

For an unknown field, users prefer to gather information as much as possible to conjecture the real situation. Unfortunately, too many messages will cause the problem of information overload and lead to improper judgments by decision makers. To combat this, DRST with its merit of using different decision makers generally provides a different "power" or "weight" to each dimension/criteria (that is, the dimension/criteria should be ordered according to decreasing or increasing preference) [33] and yields decision logics in an "if_(condition), then_(decision)" format for users to judge or examine so as to increase its real-life application [49,50]. After going through this data preprocessing procedure, the essential dimension/criterion can be identified and used to set up a formal questionnaire for COVID-19 crisis management.

**Stage 2: The mutual influence between dimensions/criteria is assessed using FDEMATEL.**

Decision-making trial and evaluation laboratory (DEMATEL), first developed by the Geneva Research Center [51], illustrates network diagrams and structural models to solve complex dynamic practical problems [52–54]. The DEMATEL technique consists of the following three processes.

Process 1: Establish a direct influence relation matrix Z. G respondents score the direct influence relation using a pairwise comparison on a scale of 0 to 4, with 0 indicating absolutely no influence and 4 indicating very strong influence, to identify the influence of criterion $i$ on criterion $j$. Each expert questionnaire forms a $n \times n$ non-negative matrix

$X^g = [x_{ij}^g]_{n \times n}$, $1 \leq g \leq G$, where $X^1$, $X^2$, ... , $X^G$ is a matrix of answers from $G$ experts based on their practical experience, and the elements of $X^g$ are expressed by $x_{ij}^g$. Therefore, the $n \times n$ average matrix Z of all experts can be established, as shown in Equation (1):

$$Z = \begin{bmatrix} z_{11} & \cdots & z_{1j} & \cdots & z_{1n} \\ \vdots & & \vdots & & \vdots \\ z_{i1} & \cdots & z_{ij} & \cdots & z_{in} \\ \vdots & & \vdots & & \vdots \\ z_{n1} & \cdots & z_{nj} & \cdots & z_{nn} \end{bmatrix} \tag{1}$$

The average score of $G$ experts is $z_{ij} = \frac{1}{G}\sum_{g=1}^{G} x_{ij}^g$. An average matrix, called the initial direct relation matrix $Z$, represents the influence of one criterion on another criterion and by another criterion.

Process 2: Normalize the direct influence matrix $D$. The normalized direct influence matrix $D$ can be obtained by normalizing the mean matrix $Z$. The matrix $D$ can be obtained using Equations (2) and (3), where the diagonals are zero.

$$D = \mu \cdot Z \tag{2}$$

$$\mu = \min\left\{ \frac{1}{\max_{1 \leq i \leq n}\sum_{j=1}^{n}|z_{ij}|}, \frac{1}{\max_{1 \leq j \leq n}\sum_{i=1}^{n}|z_{ij}|} \right\} \tag{3}$$

Process 3: Obtain the total-influence relation matrix $T$. The indirect influence continues to decline with the powers of matrix $D$, such as $D^2, D^3, \ldots, D^\infty$, such that $\lim_{q \to \infty} D^q = [0]_{n \times n}$, and $\lim_{q \to \infty}(I + D + D^2 + \ldots + D^q) = (I - D)^{-1}$, where $I$ is an $n \times n$ unit matrix. The total-influence relation matrix $T$ is an $n \times n$ matrix defined as $T = [t_{ij}]_{n \times n}, i, j = 1, 2, \ldots, n$, as shown in Equation (4).

$$T = D + D^2 + \ldots + D^q = D(I - D), \text{ when } \lim_{q \to \infty} D^q = [0]_{n \times n} \tag{4}$$

The sum of all columns and the sum of all rows form a matrix $T$, which can be obtained using Equations (5) and (6).

$$r = (r_i)_{n \times 1} = \left[\sum_{j=1}^{n} t_{ij}\right]_{n \times 1} = (r_1, \ldots, r_i, \ldots, r_n)' \tag{5}$$

$$s = (s_j)_{n \times 1} = \left[\sum_{i=1}^{n} t_{ij}\right]'_{1 \times n} = (s_1, \ldots, s_j, \ldots, s_n)' \tag{6}$$

Here, $r_i{}^D / r_i^C$ is the sum of the columns in the total-influence relation matrix $T$, representing the total influence (direct and indirect) of each criterion/dimension $i$ on all other criteria/dimensions $\left[\sum_{j=1}^{n} t_{ij}\right]_{n \times 1}$. Similarly, $s_j^D / s_j^C$ is the sum of rows in the total-influence relation matrix $T$, representing the total influence (direct and indirect) of each criterion/dimension $j$ on all other criteria/dimensions $\left[\sum_{i=1}^{n} t_{ij}\right]'_{1 \times n}$. Therefore, $j = i$, $(r_i^C + s_i^C)$ represents the relationship strength between factors—that is, $(r_i^C - s_i^C)$ indicates the degree of importance of the criterion/dimension $i$ in this system. In addition, $(r_i^C - s_i^C)$ indicates the degree of causality of total influence, representing the strength of a criterion's influence or the influence on it. If $(r_i^{C(D)} - s_i^{C(D)})$ is positive, then the criterion/dimension $i$ is a net influencer; if $(r_i^{C(D)} - s_i^{C(D)})$ is negative, then the criterion/dimension is a net receiver. The network influence relation diagram of the total-influence relation matrix $T$ can be derived by drawing the dataset $(r_i^{C(D)} + s_i^{C(D)}, r_i^{C(D)} - s_i^{C(D)})$.

**Stage 3: Incorporate FST into DEMATEL.**

Fuzzy set theory has been widely used to deal with the fuzziness of human thinking and expression in decision-making tasks. In dealing with uncertainty in decision making, an effective method called linguistic terms may be more suitable for estimation [55]. Linguistic terms can be expressed using fuzzy numbers, and triangular fuzzy numbers are the most commonly used (Opricovic and Tzeng [56]; as shown in Table 1). When users obtain decision results concerning linguistic variables (namely, fuzzy numbers), it is necessary to convert fuzzy numbers into crisp scores using defuzzification methods.

Opricovic and Tzeng [56] proposed a method to convert fuzzy data into crisp scores (CFCS), aiming at identifying left (*l*) and right (*r*) scores using fuzzy minimization and fuzzy maximization functions, with the total score determined by the weighted average method. In order to capture the fuzziness of human evaluation, the linguistic variable of "influence" is used in five influence terms, such as {no, weak, medium, strong, very strong}, which are described using triangular fuzzy numbers $(l_{ij}, m_{ij}, r_{ij})$, as shown in Table 2. Based on linguistic measures from experts, the fuzzy direct impact matrix $\widetilde{Z}$ is

$$\widetilde{Z} = [\widetilde{z}_{ij}]_{n \times n}, \text{where } \widetilde{z}_{ij} = (z^l_{ij}, z^m_{ij}, z^r_{ij}) \tag{7}$$

**Table 2.** The linguistic scale for the influence of criteria (Opricovic and Tzeng [56]).

| Linguistic Term | Triangular Fuzzy Numbers |
|---|---|
| No influence | [0, 0.1, 0.3] |
| Weak influence | [0.1, 0.3, 0.5] |
| Medium influence | [0.3, 0.5, 0.7] |
| Strong influence | [0.5, 0.7, 0.9] |
| Very strong influence | [0.7, 0.9, 1] |

From the fuzzy direct influence matrix, the normalized fuzzy direct influence matrix can be derived:

$$\widetilde{D} = \widetilde{Z}/u, \text{where } u$$
$$= \max_{i,j} \{\max_i \sum_{j-1}^{n} z_{ij}, \max_j \sum_{j-1}^{n} z_{ij}\}, \quad i, j \in \{1, \ldots, n\} \tag{8}$$
$$\widetilde{D} = [\widetilde{e}_{ij}]_{n \times n}, \quad \widetilde{e}_{ij} = (e^l_{ij}, e^m_{ij}, e^r_{ij})$$

The normalized fuzzy direct influence matrix $\widetilde{D} = (D^l, D^m, D^r)$, where $D^l = [e^l_{ij}]_{n \times n}$, $D^m = [e^m_{ij}]_{n \times n}$, and $D^r = [e^r_{ij}]_{n \times n}$. If the unit matrix (*I*) is considered, then the total fuzzy influence matrix can be obtained ($\widetilde{T}$).

$$\widetilde{T} = [\widetilde{t}_{ij}]_{n \times n}, \text{where } \widetilde{t}_{ij} = (t^l_{ij}, t^m_{ij}, t^r_{ij}) \tag{9}$$

where $$T^l = [t^l_{ij}]_{n \times n} = D^l(I - D^l)^{-1}, T^m = [t^m_{ij}]_{n \times n} = D^m(I - D^m)^{-1},$$
$$\text{and } T^r = [t^r_{ij}]_{n \times n} = D^r(I - D^r)^{-1}.$$

The total fuzzy influence matrix $\widetilde{T} = [\widetilde{t}_{ij}]_{n \times n}$ can be converted (namely, defuzzified) into the total crisp influence matrix $T = [t_{ij}]_{n \times n}$ via CFCS adoption.

## 4. Research Design and Experimental Results

This section describes the process of the questionnaire design and data collection and provides the results of the empirical analysis based on respondents' opinions on the COVID-19 crisis management evaluation architecture by implementing a fusion decision framework.

### 4.1. Questionnaire Design and Data Collection

The questionnaire design consisted of three main steps in this study. In step 1, based on an in-depth literature review and through detailed evaluations, discussions, and field experts' professional judgments, the collected data were summarized and expressed as a hierarchical structure. We set up four dimensions and 34 criteria (namely, a preliminary questionnaire) (as shown in Table 1). Satty [57] argued that too many criteria for each dimension will reduce users' decision quality—that is, a limited number of criteria for each dimension will lead to consistent results of pairwise comparisons. In order to achieve this goal, the original 33 criteria needed to be reduced.

In stage 2, preliminary questionnaires were issued to four CEOs, six general managers, and six crisis management researchers of listed companies (including 34 criteria in four dimensions). All respondents were invited to score the criteria of preliminary questionnaires and COVID-19 crisis management evaluation architecture on a scale of 0~10, with scores from high to low indicating the importance. Due to different working experiences and knowledge of experts, they clearly focused on different points of COVID-19 crisis management—that is, not all the experts targeted the same criteria and dimensions.

In order to avoid experts' bias and subjective opinions, this study analyzed all the information using DRST (as a data-driven technology). Before implementing DRST (it belongs to the group of supervised classifiers), the decision variable needs to be decided in advance. In accordance with Thangavel et al. [58], the clustering algorithm can be adopted to determine the decision variable. Generally speaking, the clusters that are bounded rely on some guideline of similarity or intrinsic characteristic such that instances in a similar cluster are alike while instances from dissimilar clusters are dissimilar [16]. The K-means approach is the simplest and most widely adopted clustering algorithm that iteratively groups the instance that is close to the center point and calculates the mean of each cluster as the centroid. However, its grouping guideline resides in the original data space, which is incapable of effectively depicting the intrinsic characteristics of the data.

To combat the above issue, one may consider the extreme learning machine clustering algorithm (ELMC) [59], which can preserve the data's intrinsic structure in an ambient feature space by considering manifold regularization [60] and avoid overfitting by adopting the Laplacian norm so as to achieve an effective and smooth outcome. The essential issue is how to determine the suitable number of clusters (i.e., K). A trial-and-error approach was thus taken.

We set K from 1 to 5, and the aggregation of the two models' forecasting accuracy was adopted as a judging measure. Five-fold cross-validation was executed to prevent the overfitting problem from occurring. According to the experiments (see Table 3), K was set to 3 to reach the optimal performance (that is, when K was set to 3, the value of the aggregated outcome was the highest). Table 4 (the results from the cluster number set to 3) shows the selected essential criteria for establishing the finalized questionnaire.

**Table 3.** The experimental results of two clustering algorithms (avg. accuracy).

| Number of Clusters | Clustering Algorithm | Classification: DRST | Aggregated Outcome (Clustering Algorithm + Classification) |
|---|---|---|---|
| K = 2 | K-means (78.5) | 81.3 | 159.8 |
|  | ELMC (83.8) | 84.6 | 168.4 |
| K = 3 | K-means (82.3) | 84.1 | 166.4 |
|  | *ELMC (86.7)* | *88.9* | *175.6* |
| K = 4 | K-means (75.3) | 77.5 | 152.5 |
|  | ELMC (80.1) | 81.4 | 161.5 |
| K = 5 | K-means (72.4) | 74.3 | 146.7 |
|  | ELMC (77.4) | 75.2 | 152.6 |

**Table 4.** The results of essential criteria selected using ELMC + DRST and K = 3.

| Dimension/Criterion | Description | Reference(s) |
|---|---|---|
| **Financial Sustainability (A)** | | |
| Controlling fixed costs ($a_1$) | Maintaining stable fixed costs and enhancing productivity | [41,45] |
| Developing as many financial resources as possible ($a_2$) | Developing as many financial resources as possible and increasing sales | [14] |
| Maintaining a good relationship with banks ($a_3$) | Maintaining a relationship with banks to meet future capital needs | [46] |
| Maintaining fixed reserves ($a_4$) | Maintaining fixed reserves and investing in own businesses | [46] |
| **Customer and Stakeholders (B)** | | |
| Increasing the delivery speed ($b_1$) | Comparing delivery schedules of actual and planned customer orders | [44,46] |
| Enhancing product functions ($b_2$) | Carrying out product performance analysis in the peer market | [45] |
| Strengthening advertising promotion ($b_3$) | Introducing advantages and effects of products and promoting brand images via media | [36] |
| Managing customer relationships ($b_4$) | Managing customer relationships and analyzing services and customer values | [41] |
| **Internal Business Process (C)** | | |
| Strengthening inventory management ($c_1$) | Optimal raw material quantity and control of products in production and finished products | [36] |
| Deploying emergency decision-making teams ($c_2$) | Deploying emergency decision-making teams to improve decision-making efficiency | [45,46] |
| Completely importing activity value management ($c_3$) | Completely importing activity value management to understand activity values | [41,44] |
| **Enablers' Learning and Growth (D)** | | |
| Motivating employees to have diversified skills ($d_1$) | Motivating employees to have diversified skills in learning organizations | [46] |
| Investing in educational training ($d_2$) | Providing internal information technology training for employees and implementing incentive mechanisms in the mode of teamwork | [36,41] |
| Designing complete performance reward systems ($d_3$) | Designing complete performance reward systems to motivate employees | [45,46] |

The finalized questionnaires were sent to 8 CEOs and 14 general managers from industrial areas and 10 crisis management researchers from academic areas in China's first-tier cities. The 8 CEOs and 14 general managers were familiar with crisis management and had at least 12 years of working experience. The crisis management researchers were university professors specializing in risk analysis and crisis management. The questionnaire survey was conducted from December 2020 to April 2021 through 60 to 90 min of online/offline interviews. Considering the interviewees' responses, the relationship between any two criteria was obtained via pairwise comparison, and the direct influence evaluation was generated on a five-point scale from 0 ("absolutely no influence") to 4 ("very strong influence"). Finally, 34 valid questionnaires were imported into the FDEMATEL model as the basis of empirical analysis.

*4.2. Influence Relation Matrix (INRM) Creation by FDEMATEL*

The influence relation matrix (INRM) can be derived from FDEMATEL based on the expert questionnaire survey. The 34 experts with crisis management backgrounds are invited to score the dependence of each criterion on the others and to obtain the initial influence matrix $\widetilde{Z}$ via pairwise comparison. The initial influence relation matrix $\widetilde{Z}$ was normalized using Equation (8) to determine the direct influence relation matrix $\widetilde{D}$.

Equation (9) was used to derive the fuzzy total impact relationship matrix $\widetilde{T}$ (as shown in Tables 5–8) to identify INRM.

**Table 5.** Fuzzy normalized direct and indirect influence relation matrix $T^l$ (left).

| Criterion | $a_1$ | $a_2$ | $a_3$ | $a_4$ | $b_1$ | $b_2$ | $b_3$ | $b_4$ | $c_1$ | $c_2$ | $c_3$ | $d_1$ | $d_2$ | $d_3$ |
|---|---|---|---|---|---|---|---|---|---|---|---|---|---|---|
| $a_1$ | 0.013 | 0.065 | 0.017 | 0.025 | 0.047 | 0.053 | 0.076 | 0.025 | 0.011 | 0.036 | 0.060 | 0.025 | 0.048 | 0.006 |
| $a_2$ | 0.019 | 0.010 | 0.011 | 0.015 | 0.026 | 0.025 | 0.028 | 0.009 | 0.012 | 0.015 | 0.040 | 0.014 | 0.013 | 0.004 |
| $a_3$ | 0.064 | 0.067 | 0.012 | 0.050 | 0.079 | 0.043 | 0.063 | 0.039 | 0.044 | 0.058 | 0.084 | 0.051 | 0.042 | 0.055 |
| $a_4$ | 0.085 | 0.096 | 0.073 | 0.015 | 0.090 | 0.086 | 0.085 | 0.051 | 0.031 | 0.051 | 0.070 | 0.031 | 0.036 | 0.034 |
| $b_1$ | 0.037 | 0.064 | 0.019 | 0.022 | 0.015 | 0.061 | 0.044 | 0.032 | 0.027 | 0.039 | 0.045 | 0.021 | 0.036 | 0.017 |
| $b_2$ | 0.021 | 0.052 | 0.023 | 0.022 | 0.050 | 0.013 | 0.058 | 0.061 | 0.018 | 0.036 | 0.062 | 0.014 | 0.031 | 0.006 |
| $b_3$ | 0.061 | 0.072 | 0.037 | 0.041 | 0.053 | 0.035 | 0.021 | 0.026 | 0.028 | 0.068 | 0.064 | 0.020 | 0.060 | 0.011 |
| $b_4$ | 0.075 | 0.088 | 0.041 | 0.042 | 0.079 | 0.069 | 0.081 | 0.015 | 0.031 | 0.062 | 0.048 | 0.027 | 0.046 | 0.020 |
| $c_1$ | 0.032 | 0.051 | 0.012 | 0.019 | 0.036 | 0.037 | 0.045 | 0.030 | 0.010 | 0.074 | 0.079 | 0.032 | 0.040 | 0.013 |
| $c_2$ | 0.041 | 0.064 | 0.034 | 0.033 | 0.055 | 0.055 | 0.071 | 0.051 | 0.060 | 0.020 | 0.084 | 0.024 | 0.040 | 0.010 |
| $c_3$ | 0.027 | 0.043 | 0.015 | 0.020 | 0.044 | 0.013 | 0.042 | 0.013 | 0.017 | 0.033 | 0.012 | 0.010 | 0.015 | 0.005 |
| $d_1$ | 0.014 | 0.052 | 0.010 | 0.014 | 0.031 | 0.025 | 0.039 | 0.025 | 0.049 | 0.057 | 0.060 | 0.006 | 0.018 | 0.023 |
| $d_2$ | 0.019 | 0.044 | 0.007 | 0.005 | 0.014 | 0.013 | 0.015 | 0.011 | 0.019 | 0.021 | 0.023 | 0.018 | 0.004 | 0.004 |
| $d_3$ | 0.021 | 0.019 | 0.020 | 0.005 | 0.027 | 0.022 | 0.022 | 0.006 | 0.035 | 0.038 | 0.041 | 0.008 | 0.010 | 0.002 |

**Table 6.** Fuzzy normalized direct and indirect influence relation matrix $T^m$ (medium).

| Criterion | $a_1$ | $a_2$ | $a_3$ | $a_4$ | $b_1$ | $b_2$ | $b_3$ | $b_4$ | $c_1$ | $c_2$ | $c_3$ | $d_1$ | $d_2$ | $d_3$ |
|---|---|---|---|---|---|---|---|---|---|---|---|---|---|---|
| $a_1$ | 0.062 | 0.134 | 0.063 | 0.071 | 0.109 | 0.109 | 0.141 | 0.075 | 0.057 | 0.096 | 0.128 | 0.072 | 0.101 | 0.042 |
| $a_2$ | 0.063 | 0.056 | 0.046 | 0.050 | 0.071 | 0.070 | 0.081 | 0.046 | 0.053 | 0.063 | 0.095 | 0.053 | 0.056 | 0.033 |
| $a_3$ | 0.133 | 0.148 | 0.060 | 0.107 | 0.153 | 0.110 | 0.140 | 0.098 | 0.104 | 0.129 | 0.164 | 0.106 | 0.104 | 0.103 |
| $a_4$ | 0.156 | 0.180 | 0.132 | 0.066 | 0.167 | 0.158 | 0.165 | 0.111 | 0.092 | 0.123 | 0.154 | 0.089 | 0.100 | 0.085 |
| $b_1$ | 0.093 | 0.132 | 0.062 | 0.070 | 0.069 | 0.119 | 0.109 | 0.082 | 0.078 | 0.098 | 0.113 | 0.068 | 0.089 | 0.060 |
| $b_2$ | 0.078 | 0.119 | 0.069 | 0.067 | 0.113 | 0.062 | 0.123 | 0.113 | 0.070 | 0.095 | 0.129 | 0.061 | 0.086 | 0.042 |
| $b_3$ | 0.124 | 0.147 | 0.089 | 0.094 | 0.121 | 0.097 | 0.083 | 0.081 | 0.085 | 0.135 | 0.138 | 0.072 | 0.118 | 0.051 |
| $b_4$ | 0.142 | 0.168 | 0.096 | 0.097 | 0.152 | 0.137 | 0.157 | 0.066 | 0.091 | 0.133 | 0.124 | 0.081 | 0.108 | 0.069 |
| $c_1$ | 0.092 | 0.121 | 0.056 | 0.068 | 0.100 | 0.097 | 0.112 | 0.080 | 0.056 | 0.136 | 0.148 | 0.080 | 0.094 | 0.057 |
| $c_2$ | 0.105 | 0.139 | 0.084 | 0.082 | 0.125 | 0.119 | 0.143 | 0.108 | 0.118 | 0.079 | 0.159 | 0.076 | 0.099 | 0.052 |
| $c_3$ | 0.076 | 0.103 | 0.054 | 0.061 | 0.098 | 0.059 | 0.098 | 0.053 | 0.061 | 0.086 | 0.062 | 0.046 | 0.061 | 0.036 |
| $d_1$ | 0.064 | 0.114 | 0.051 | 0.059 | 0.088 | 0.079 | 0.099 | 0.070 | 0.097 | 0.112 | 0.122 | 0.043 | 0.070 | 0.064 |
| $d_2$ | 0.063 | 0.096 | 0.040 | 0.037 | 0.057 | 0.054 | 0.064 | 0.051 | 0.059 | 0.067 | 0.074 | 0.055 | 0.039 | 0.031 |
| $d_3$ | 0.061 | 0.065 | 0.052 | 0.035 | 0.070 | 0.062 | 0.065 | 0.038 | 0.075 | 0.084 | 0.091 | 0.042 | 0.048 | 0.028 |

**Table 7.** Fuzzy normalized direct and indirect influence relation matrix $T^r$ (right).

| Criterion | $a_1$ | $a_2$ | $a_3$ | $a_4$ | $b_1$ | $b_2$ | $b_3$ | $b_4$ | $c_1$ | $c_2$ | $c_3$ | $d_1$ | $d_2$ | $d_3$ |
|---|---|---|---|---|---|---|---|---|---|---|---|---|---|---|
| $a_1$ | 0.262 | 0.364 | 0.234 | 0.243 | 0.319 | 0.307 | 0.355 | 0.257 | 0.243 | 0.306 | 0.353 | 0.246 | 0.293 | 0.199 |
| $a_2$ | 0.229 | 0.248 | 0.188 | 0.195 | 0.246 | 0.238 | 0.265 | 0.197 | 0.207 | 0.238 | 0.282 | 0.197 | 0.215 | 0.163 |
| $a_3$ | 0.363 | 0.414 | 0.259 | 0.310 | 0.392 | 0.344 | 0.395 | 0.310 | 0.319 | 0.369 | 0.424 | 0.304 | 0.328 | 0.280 |
| $a_4$ | 0.382 | 0.442 | 0.327 | 0.271 | 0.407 | 0.386 | 0.419 | 0.322 | 0.312 | 0.370 | 0.420 | 0.294 | 0.326 | 0.270 |
| $b_1$ | 0.296 | 0.362 | 0.235 | 0.246 | 0.283 | 0.322 | 0.333 | 0.266 | 0.266 | 0.311 | 0.345 | 0.244 | 0.283 | 0.218 |
| $b_2$ | 0.276 | 0.346 | 0.240 | 0.240 | 0.321 | 0.263 | 0.340 | 0.290 | 0.255 | 0.304 | 0.353 | 0.233 | 0.276 | 0.198 |
| $b_3$ | 0.337 | 0.392 | 0.274 | 0.279 | 0.348 | 0.314 | 0.321 | 0.277 | 0.285 | 0.358 | 0.382 | 0.258 | 0.319 | 0.219 |
| $b_4$ | 0.364 | 0.423 | 0.287 | 0.295 | 0.387 | 0.365 | 0.403 | 0.274 | 0.303 | 0.369 | 0.384 | 0.278 | 0.327 | 0.248 |
| $c_1$ | 0.292 | 0.352 | 0.229 | 0.244 | 0.312 | 0.299 | 0.333 | 0.264 | 0.243 | 0.342 | 0.372 | 0.256 | 0.288 | 0.216 |
| $c_2$ | 0.321 | 0.390 | 0.269 | 0.271 | 0.356 | 0.338 | 0.382 | 0.303 | 0.319 | 0.309 | 0.401 | 0.266 | 0.309 | 0.223 |
| $c_3$ | 0.248 | 0.301 | 0.203 | 0.213 | 0.280 | 0.235 | 0.289 | 0.212 | 0.223 | 0.268 | 0.262 | 0.197 | 0.228 | 0.172 |
| $d_1$ | 0.254 | 0.333 | 0.215 | 0.225 | 0.291 | 0.272 | 0.310 | 0.244 | 0.270 | 0.309 | 0.337 | 0.208 | 0.253 | 0.214 |
| $d_2$ | 0.227 | 0.286 | 0.180 | 0.180 | 0.232 | 0.219 | 0.246 | 0.200 | 0.211 | 0.240 | 0.262 | 0.197 | 0.196 | 0.159 |
| $d_3$ | 0.221 | 0.252 | 0.188 | 0.176 | 0.240 | 0.223 | 0.243 | 0.185 | 0.223 | 0.252 | 0.274 | 0.182 | 0.203 | 0.155 |

**Table 8.** Fuzzy normalized direct and indirect influence relation matrix *T* for the criteria (average).

| Criterion | $a_1$ | $a_2$ | $a_3$ | $a_4$ | $b_1$ | $b_2$ | $b_3$ | $b_4$ | $c_1$ | $c_2$ | $c_3$ | $d_1$ | $d_2$ | $d_3$ |
|---|---|---|---|---|---|---|---|---|---|---|---|---|---|---|
| $a_1$ | 0.112 | 0.188 | 0.105 | 0.113 | 0.158 | 0.156 | 0.190 | 0.119 | 0.103 | 0.146 | 0.180 | 0.114 | 0.147 | 0.082 |
| $a_2$ | 0.103 | 0.105 | 0.082 | 0.087 | 0.114 | 0.111 | 0.125 | 0.084 | 0.091 | 0.105 | 0.139 | 0.088 | 0.095 | 0.066 |
| $a_3$ | 0.187 | 0.210 | 0.111 | 0.156 | 0.208 | 0.166 | 0.200 | 0.149 | 0.156 | 0.186 | 0.224 | 0.154 | 0.158 | 0.146 |
| $a_4$ | 0.208 | 0.239 | 0.177 | 0.117 | 0.221 | 0.210 | 0.223 | 0.162 | 0.145 | 0.182 | 0.215 | 0.138 | 0.154 | 0.129 |
| $b_1$ | 0.142 | 0.186 | 0.105 | 0.113 | 0.122 | 0.168 | 0.162 | 0.127 | 0.123 | 0.150 | 0.168 | 0.111 | 0.136 | 0.098 |
| $b_2$ | 0.125 | 0.172 | 0.111 | 0.110 | 0.161 | 0.113 | 0.174 | 0.155 | 0.114 | 0.145 | 0.181 | 0.103 | 0.131 | 0.082 |
| $b_3$ | 0.174 | 0.204 | 0.133 | 0.138 | 0.174 | 0.149 | 0.142 | 0.128 | 0.132 | 0.187 | 0.195 | 0.117 | 0.166 | 0.094 |
| $b_4$ | 0.194 | 0.227 | 0.141 | 0.144 | 0.206 | 0.191 | 0.214 | 0.118 | 0.142 | 0.188 | 0.185 | 0.129 | 0.160 | 0.112 |
| $c_1$ | 0.139 | 0.174 | 0.099 | 0.110 | 0.149 | 0.144 | 0.163 | 0.124 | 0.103 | 0.184 | 0.200 | 0.123 | 0.141 | 0.095 |
| $c_2$ | 0.156 | 0.198 | 0.129 | 0.129 | 0.179 | 0.171 | 0.199 | 0.154 | 0.166 | 0.136 | 0.215 | 0.122 | 0.150 | 0.095 |
| $c_3$ | 0.117 | 0.149 | 0.090 | 0.098 | 0.141 | 0.102 | 0.143 | 0.093 | 0.100 | 0.129 | 0.112 | 0.084 | 0.101 | 0.071 |
| $d_1$ | 0.111 | 0.167 | 0.092 | 0.099 | 0.137 | 0.125 | 0.149 | 0.113 | 0.139 | 0.159 | 0.173 | 0.086 | 0.114 | 0.101 |
| $d_2$ | 0.103 | 0.142 | 0.076 | 0.074 | 0.101 | 0.095 | 0.108 | 0.087 | 0.096 | 0.109 | 0.120 | 0.090 | 0.080 | 0.065 |
| $d_3$ | 0.101 | 0.112 | 0.087 | 0.072 | 0.112 | 0.102 | 0.110 | 0.076 | 0.111 | 0.124 | 0.135 | 0.077 | 0.087 | 0.062 |

We calculated $(r_i + s_i)$ and $(r_i - s_i)$ using the FDEMATEL technique, as shown in Table 9. According to Table 9, among the four dimensions, dimension A (financial sustainability (A)) had the greatest influence ($r_i - s_i = 0.054$), and dimension C (internal business process (C)) and dimension D (enablers' learning and growth (D)) were influenced by other dimensions. Here, the $(r_i - s_i)$ of each component was calculated as a driver (positive value) or a receiver (negative value) and indicated the determinants influencing other components or being influenced by other components.

**Table 9.** Sum of cause $r_i$ and effect $s_i$ influences among the core dimensions and criteria.

| Dimension/Criterion | Row Sum ($r_i$) | Column Sum ($s_i$) | $r_i + s_i$ | $r_i - s_i$ |
|---|---|---|---|---|
| **Financial Sustainability (A)** | **0.584** | **0.530** | **1.114** | **0.054** |
| Controlling fixed costs ($a_1$) | 0.518 | 0.610 | 1.128 | −0.092 |
| Developing as many financial resources as possible ($a_2$) | 0.376 | 0.741 | 1.117 | −0.365 |
| Maintaining a good relationship with banks ($a_3$) | 0.663 | 0.474 | 1.137 | 0.188 |
| Maintaining fixed reserves ($a_4$) | 0.742 | 0.473 | 1.214 | 0.269 |
| **Customer and Stakeholders (B)** | **0.587** | **0.575** | **1.162** | **0.011** |
| Increasing the delivery speed ($b_1$) | 0.578 | 0.664 | 1.242 | −0.086 |
| Enhancing product functions ($b_2$) | 0.602 | 0.620 | 1.222 | −0.017 |
| Strengthening advertising promotion ($b_3$) | 0.593 | 0.691 | 1.283 | −0.098 |
| Managing customer relationships ($b_4$) | 0.729 | 0.528 | 1.257 | 0.201 |
| **Internal Business Process (C)** | **0.538** | **0.594** | **1.132** | **−0.056** |
| Strengthening inventory management ($c_1$) | 0.298 | 0.405 | 0.703 | −0.107 |
| Deploying emergency decision-making teams ($c_2$) | 0.373 | 0.260 | 0.633 | 0.113 |
| Completely importing activity value management ($c_3$) | 0.290 | 0.295 | 0.585 | −0.006 |
| **Enablers' Learning and Growth (D)** | **0.427** | **0.436** | **0.863** | **−0.009** |
| Motivating employees to have diversified skills ($d_1$) | 0.300 | 0.253 | 0.553 | 0.048 |
| Investing in educational training ($d_2$) | 0.234 | 0.281 | 0.515 | −0.046 |
| Designing complete performance reward systems ($d_3$) | 0.226 | 0.227 | 0.453 | −0.001 |

Among the four dimensions, dimension A was the most influential factor, which indicated that it had the greatest influence and its improvement will lead to the improvement of other dimensions. Financial sustainability (A) ($r_i - s_i = 0.047$) and customer and stakeholders (B) ($r_i - s_i = 0.011$) were positive, indicating that they acted directly on other dimensions as driving factors. On the other hand, internal business process (C) ($r_i - s_i = -0.056$) and enablers' learning and growth (D) ($r_i - s_i = -0.009$) were negative, indicating that these dimensions were influenced by other dimensions as receivers. Comparing the 14 criteria, the criterion of "maintaining fixed reserves ($a_4$)" showed the maximum ($r_i - s_i$) of 0.269, indicating the greatest influence of this criterion on other

criteria. However, the criterion of "developing as many financial resources as possible ($a_2$)" scored the lowest, indicating that this indicator was the most easily influenced by other criteria.

This study used the FDEMATEL technique to measure the degree of interaction between 4 dimensions and 14 criteria and to construct INRM of dimensions and criteria. In Figure 1 the horizontal axis shows the total relation among variables, and the vertical axis shows the causality between variables ($r_i - s_i$). INRM can help us to clearly understand the interdependence between the dimensions in the COVID-19 crisis management evaluation architecture.

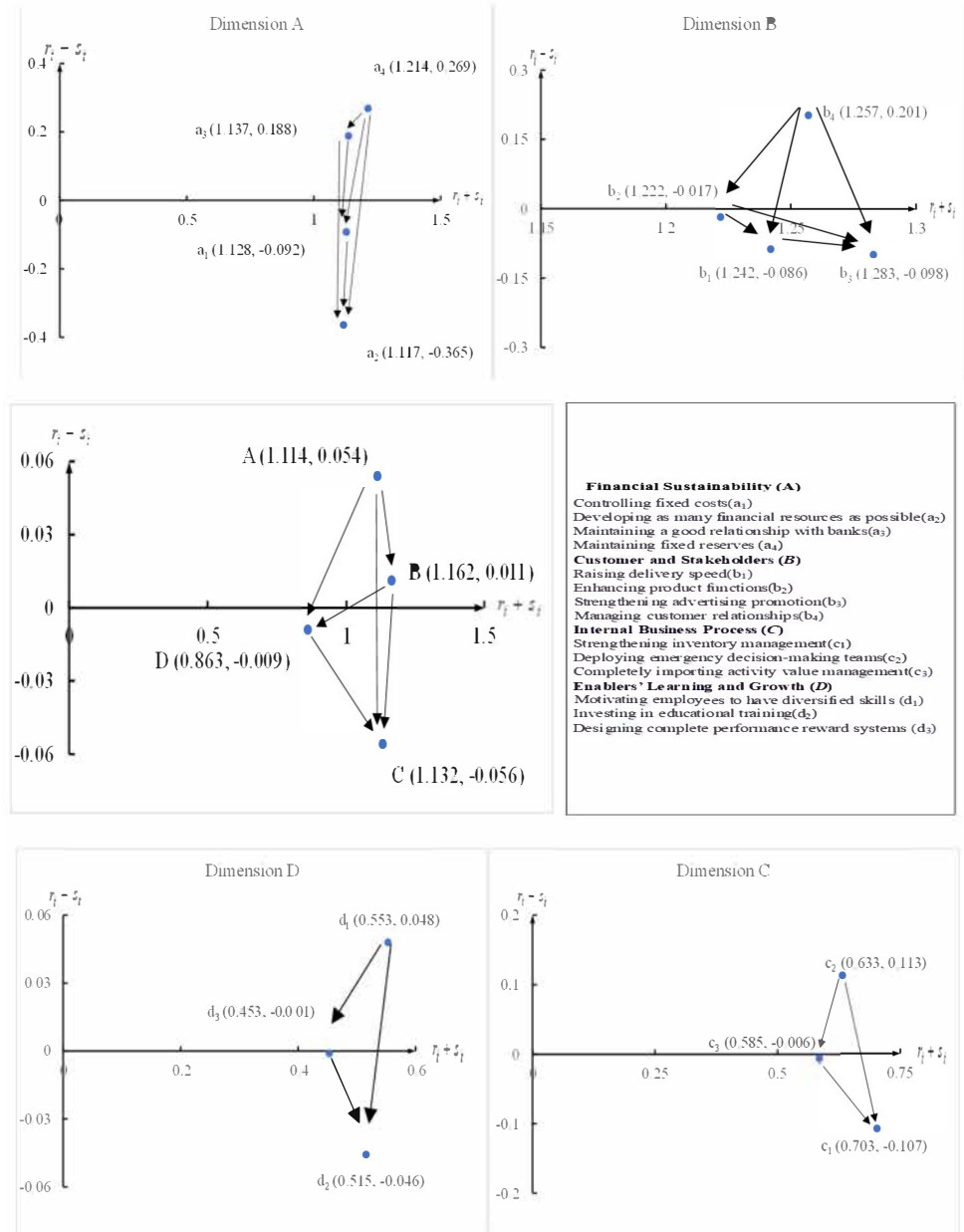

**Figure 1.** The INRM results of the influence relationships based on FDEMATEL.

As shown in Figure 1, dimension A ((financial sustainability (A)) confirmed its direct influences on other dimensions, and dimension B (customer and stakeholders (B)) influenced dimension C (internal business process (C)) and dimension D (enablers' learning and growth (D)). Hence, according to the results, financial sustainability, as well as customer and stakeholders, were the determinants for the sustainable development of enterprises.

Dimensions C and D were below the horizontal axis, indicating that they were influenced (non-causal) dimensions.

The analysis of the direct network influence relationship between the criteria in the dimensions ran as follows. The criteria of ($a_4$) (maintaining fixed reserves), ($b_4$) (managing customer relationships), ($c_2$) (deploying emergency decision-making teams), and ($d_1$) (motivating employees to have diversified skills) had significant influences in some dimensions. The aforementioned four criteria were the cores of their dimensions and had great influences on their dimensions. As a result, INRM gave us a clearer picture of the interdependence between the criteria in the COVID-19 crisis management evaluation architecture.

## 5. Discussion and Implication

This study combined the ELMC and DRST technologies and the FDEMATEL method to provide a practical framework for enterprises to evaluate and enhance their crisis management following the COVID-19 outbreak in 2019. Figure 1 illustrates the causal relationships between the systems (dimensions) and sub-systems (criteria) for assessing COVID-19 crisis management. According to the degree of relationship, the priority of the improved dimensions was financial sustainability (A), customer and stakeholders (B), enablers' learning and growth (D), and internal business process (C). The results showed that financial sustainability (A) had the most important and direct influence on other dimensions.

Financial sustainability (A), if selected as the priority to be improved, had a cumulative effect on solving problems. COVID-19 has had significant influences on the business environment and enterprise operations. As enterprises have faced various challenges due to the pandemic, their operating cash flow may be reduced because of government requirements regarding controlling the flow of people. The shortage of upstream raw materials may also cause a failure to fill customer orders in a timely fashion. Furthermore, changes in the consumption models of consumers may influence corporate revenues as well. When enterprises are under the double pressure of collapse in orders and shortage of raw materials, the operating cash flow gradually fails to meet fixed operating cash expenditures, let alone leaving enough funds to repay maturing loans [61,62].

If corporate managers discover any of the above financial problems, then in order to maintain financial sustainability, the following conditions must be reviewed as soon as possible [63,64]. First, based on cost-effectiveness analysis, unnecessary expenses can be reduced or postponed without influencing operations during an emergency period. Second, for asset reconstruction, non-urgent assets (such as idle land or plants) can be disposed of to raise cash levels. Third, when under pressure to produce financial reconstruction, the response methods can be assessed according to corporate conditions, such as issuing preferred stock and debt for equity swaps to reduce the debt ratio. Fourth, for contract reconstruction, enterprises should negotiate with creditors, such as financial institutions, to reduce interest burdens or extend the debt period and negotiate with counterparties to postpone payment or delivery so as to get through tough times together. The pandemic has caused a sharp drop in revenues and made "Cash is King" relevant again. Appropriate cash reserves can allow enterprises to absorb harsh financial disturbances [65]. It is a reasonable prevention measure to prepare for rainy days on sunny days.

According to the causality diagram (Figure 1), the top-priority key criterion of maintaining fixed reserves ($a_4$) within the financial sustainability (A) dimension must be carried out. It is suggested that enterprises make plans to accumulate crisis reserves and properly use them so as to provide cash liquidity for operations in the event of crises. By examining financial liquidity and financing capacities, such as short-term cash flow evaluation, enterprises can then analyze their cost reduction measures, develop contingency financial plans, and invest capital for important businesses and services in crises so that capital can create value more effectively.

For the second key criterion of financial sustainability (A), which was maintaining a good relationship with banks ($a_3$), enterprises' investment pipeline should be kept stable. Enterprises should also actively develop as many financial resources as possible, such as applying for government subsidies. Moreover, enterprises should change their business models to cope with a crisis and take advantage of opportunities to reallocate their capital. Finally, capital shall be invested in businesses that increase turnover, strengthen the foundation of enterprises, and enhance sales. Enterprises should move toward the goal of improving productivity, reducing costs, advancing productivity, and creating more added value.

In terms of the top-priority key criterion of customer and stakeholders (B), which was managing customer relationships ($b_4$), enterprises should invest resources in customer relationship management and consider customers as important corporate assets. They should aim to improve customer satisfaction and loyalty. Enterprises performing well in customer relationship management will be less hurt in the face of black swan events [65]. It is suggested that enterprises manage customer relationships before the pre-crisis prevention and preparation stage to reduce any damages encountered in the face of future crises by virtue of high customer loyalty.

In terms of enablers' learning and growth (D), its top-priority key criterion was motivating employees to have diversified skills ($d_1$). Thus, enterprises can adopt interdepartmental reward systems. During the continuous interdepartmental training process, employees with secondary expertise can be rewarded and promoted to different areas depending on the circumstances to develop interdepartmental talents. When a crisis breaks out, enterprises typically reduce their manpower, pushing employees to provide interdepartmental support. If the enterprises have already prepared themselves with interdepartmental talents, then this lower costs and ensures their normal operations.

Finally, the top-priority key criterion of the internal business process (C) was deploying emergency decision-making teams ($c_2$). During a pandemic crisis, corporate plans usually cannot keep up with the rapid changes in the market, and every minute counts when a crisis breaks out. By improving the speed of internal decision-making, the chances for enterprises to survive can increase. Hence, enterprises must first design and prepare their internal decision-making process and set up emergency decision-making teams. Those emergency decision-making teams should pay attention to their enterprise's short-term, medium-term, and long-term strategies and adjust the corporate planning and strategy according to the situation. When a crisis breaks out, enterprises can then smoothly and quickly release policies and deploy resources in advance so as to reduce the damage caused by the crisis.

## 6. Conclusions

The COVID-19 pandemic has caused tremendous financial troubles and disturbances to enterprises globally, and even top-ranked corporations are still having trouble regaining stability after encountering various detrimental financial impacts. Most enterprises have been forced to reduce production capacity, lay off employees, and even worse, shut down their operations in response to this unprecedented financial pressure. In order to guide a suitable direction for managers to follow and recover sooner from this financial crisis, the study conducted an in-depth literature review to develop an advanced crisis management architecture via a fusion model integration that can assist them in realizing which dimension/criterion poses considerable influence and prioritizing resource allocation strategies without any time delay and waste so as to quickly react to market fluctuations.

The results showed that the ranking priority for improvement between dimensions was financial sustainability (A), customer and stakeholders (B), enablers' learning and growth (D), and internal business process (C). The criterion of "maintaining fixed reserves ($a_4$)" was confirmed to have the maximum influence on the other criteria, meaning it should be the first improvement objective among the criteria. According to the above results, financial sustainability is the most important dimension. Therefore, it is suggested

that enterprises establish a mechanism to improve their financial resilience; set up a contingency plan, which can be carried out through a pressure test and a real-time cash flow prediction mechanism in the financial aspect; and rapidly examine the corporate liquidity and financing capacity. The specific practical plan goes as follows:

➢　Prepare a short-term cash flow forecast on a weekly basis for 3–4 months in the future and review and update the forecast according to actual conditions when they occur.
➢　Analyze factors that may influence short-term cash flow results to help recognize possible risks.
➢　Define ways to quickly reduce costs and preserve cash.
➢　Pay attention to the influence of cash preservation measures on cash flow during the weekly review.
➢　Use the short-term cash flow forecast as a convincing quantitative tool in negotiations with financiers and investors.
➢　Make a comprehensive inventory of mortgages/pledge of assets and existing financing amounts.
➢　Use existing financing amounts to maximize available cash wherever possible.
➢　Analyze major fixed cash expenditure items.

With the end of COVID-19 still far away and in the face of major and rapid economic changes, the strategy for enterprises to survive financially has become critical in maintaining and strengthening their financial resilience. By seeking sustainable financial goals, enterprises can also achieve sustainability.

In addition, in the face of different crises, companies can use the four dimensions of the BSCs to help companies focus more smoothly on the strategic goals they want to achieve. Before a crisis breaks out, being prepared for the crisis can most effectively reduce the losses caused by the crisis, and maintaining financial discipline is a good strategy for the enterprise. Maintaining a crisis preparedness fund and not investing blindly can help companies have good financial backing in the event of a crisis. When the crisis breaks out, companies can choose to engage in more diversified business models and seize this opportunity to record and analyze the value of each business well because after the crisis occurs, companies still have to pursue profits, find target customers, eliminate waste, and retain the business that allows the company to grow.

Although this study succeeded in constructing a practical crisis management model, there were still some limitations. The crisis management evaluation framework proposed in this study was based on general guidelines; however, companies should take into account the characteristics of the company's industry, industry profiles, and the differences between country conditions. They should consider the characteristics of enterprises and choose the most suitable crisis management tool. It would be worth exploring some interesting ideas in future work in order to extend this study's analysis. For example, multiple comparisons with other data selection models could show how the model in this study was superior to other approaches. Moreover, doing so could help to evaluate the appropriateness of enterprise crisis management under harsh times in the market as discussed in this study.

**Author Contributions:** Conceptualization, K.-H.H. and C.D.; methodology, S.-J.L.; validation, K.-H.H., F.-H.C. and S.-J.L.; formal analysis, C.D.; investigation, S.-J.L.; resources, F.-H.C.; data curation, C.D. and K.-H.H.; writing—original draft preparation, F.-H.C.; writing—review and editing, S.-J.L. and M.-C.H.; visualization, K.-H.H.; supervision, S.-J.L. and M.-C.H. All authors have read and agreed to the published version of the manuscript.

**Funding:** This research was funded by the Ministry of Science and Technology, Taiwan, grant numbers "110-2410-H-034-011" and "109-2410-H-034-034-MY2", and the Department of Education of Guangdong Province, China, nos. 2020WTSCX139 and 2021ZDJS131.

**Data Availability Statement:** The datasets used during the current study are available from the corresponding author on reasonable request.

**Acknowledgments:** The authors would like to thank the Ministry of Science and Technology, Taiwan and the Department of Education of Guangdong Province, China for financially supporting this work.

**Conflicts of Interest:** The authors declare no conflict of interest.

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
