# Peer review of "A Fusion Decision-Making Architecture for COVID-19 Crisis Analysis and Management"

_electronics, doi:10.3390/electronics11111793_

Round 1

Reviewer 1 Report

The manuscript is devoted to the revision of the architecture of anti-crisis management of the enterprise in order to quickly recover after a collision with extremely negative economic consequences after a pandemic. The authors present a fusion architecture that combines artificial intelligence and decision-making according to several criteria. The relationships between the criteria are determined. Using this architecture allows the manager to prioritize improvement plans and direct resources to key areas. The exact priorities of improvement have been identified. Appropriate recommendations are given.

The authors have done a lot of work on the actual topic, there are signs of scientific novelty, the reasoning is logically correct and justified. I think that the article can be published.

Author Response

Dear Reviewers:

Those comments are all valuable and very helpful for revising and improving our paper, as well as the important guiding significance to our research.

We are grateful for this revision opportunity and thank you for your suggestion and for the reviewers’ comments concerning our manuscript entitled “A Fusion Decision Making Architecture for COVID-19 Crisis Analysis and Management” (ID: electronics-1742661). The comments again from the reviewers are highly appreciated, and those valuable comments did help us improve the manuscript in many aspects. We have revised and replied each comment and the major changes have been marked in Red.

The main corrections in the paper and the point-by-point responds to your and reviewer’s comments are as flowing:

Responds to the reviewer’s comments:

The authors are extremely grateful to reviewer involved for providing his/her excellent comments and valuable advice in this paper. We have done our best to provide a satisfactory revision addressing the reviewers’ concerns. Point-by-point responses to each comment are listed below and major changes to the manuscript have been marked in red.

Point-by-point responses are given as follow.

Reviewer #1

  1. The manuscript is devoted to the revision of the architecture of anti-crisis management of the enterprise in order to quickly recover after a collision with extremely negative economic consequences after a pandemic. The authors present a fusion architecture that combines artificial intelligence and decision-making according to several criteria. The relationships between the criteria are determined. Using this architecture allows the manager to prioritize improvement plans and direct resources to key areas. The exact priorities of improvement have been identified. Appropriate recommendations are given.
  2. The authors have done a lot of work on the actual topic, there are signs of scientific novelty, the reasoning is logically correct and justified. I think that the article can be published.

Response: Thank you for this comment and suggestion.

Reviewer 2 Report

The paper brings significant contributions to the field of risk analysis and crisis management and provides an architecture that can support managers to prioritize improvement plans and deploy resources to key areas.

Please find below some suggestions:

The paragraph (178): The relationships within the strategy map are as follows. It is not clear, from the next paragraphs, which are relationships mentioned before. Maybe a rephrase is useful here

The authors do not mention if there are  any limitations (s) that can impact the result of the study

I suggest the authors emphasize how the architecture developed in this paper can be applied in other types of crises not only the ones generated by Covid pandemic

Author Response

Dear Reviewers:

Those comments are all valuable and very helpful for revising and improving our paper, as well as the important guiding significance to our research.

We are grateful for this revision opportunity and thank you for your suggestion and for the reviewers’ comments concerning our manuscript entitled “A Fusion Decision Making Architecture for COVID-19 Crisis Analysis and Management” (ID: electronics-1742661). The comments again from the reviewers are highly appreciated, and those valuable comments did help us improve the manuscript in many aspects. We have revised and replied each comment and the major changes have been marked in Red.

The main corrections in the paper and the point-by-point responds to your and reviewer’s comments are as flowing:

Responds to the reviewer’s comments:

The authors are extremely grateful to reviewer involved for providing his/her excellent comments and valuable advice in this paper. We have done our best to provide a satisfactory revision addressing the reviewers’ concerns. Point-by-point responses to each comment are listed below and major changes to the manuscript have been marked in red.

Point-by-point responses are given as follow.

Reviewer #2

The paper brings significant contributions to the field of risk analysis and crisis management and provides an architecture that can support managers to prioritize improvement plans and deploy resources to key areas.

Please find below some suggestions:

  1. The paragraph (178): The relationships within the strategy map are as follows. It is not clear, from the next paragraphs, which are relationships mentioned before. Maybe a rephrase is useful here

Response: Thanks for your valuable suggestions, we have made considerable modifications in this context. (Please see 2. Research background)

2.2. Balanced scorecards (BSCs)

The concept of balanced scorecards (BSCs) was first proposed by KPMG in order to design a performance appraisal system for Apple Inc. and was later developed by Professor Robert Kaplan at Harvard University and David Norton in business circles. After summarizing the successful experience of 12 companies in developing perfor-mance management systems, Kaplan and Norton proposed and promoted BSCs to the world. After that, they published articles on BSCs in the Harvard Business Review in succession. Balanced Scored Card-Measures that Drive Performance [14] first pointed out the benefits gained by companies in using BSCs for performance appraisal. Putting the Balanced Scorecard to Work [35] then stated that the basis of performance apprais-al indicator selection is the key success factor of corporate strategies. Using the Bal-anced Scored Card as a Strategic Management System [14] next solved two problems: one is the importance of BSCs, as the book discusses in detail their importance as a strategic management tool for corporate strategic practice; another is the framework, as the book outlines and explains the framework of BSCs as a strategy and perfor-mance management tool. 

BSCs is a tool to measure company performance in four dimensions: finance, customers, employees, and internal processes of an enterprise, and a strategy map can build a framework for the strategic goals of the four dimensions of an enterprise [36]. The strategy map is an extension of the BSCs, which shows how the company trans-forms the company's different assets into the company's desired results, and the com-pany can develop its own strategy map according to its own different goals. With the four dimensions of the BSCs, a strategy map model is created for various industries, which is a reference for each enterprise to implement strategies, which can not only focus on the company strategy but also significantly enhance the cooperation and co-ordination within the company [35, 36]. Beasley et al. [37] stated that the validity and effectiveness of BSCs can be strengthened by incorporating them into enterprises’ cri-sis management. By doing that we can link crisis management to enterprises’ strategic performance evaluation as well as assist managers to target profit maximization under anticipated risk exposure and to expand the scope of crises management.

  1. The authors do not mention if there are any limitations (s) that can impact the result of the study

Response: Thanks for your valuable suggestions, we have made considerable modifications in this context. (Please see 6. Conclusion)

With the end of COVID-19 still far in sight and in the face of major and rapid economic changes, the strategy for enterprises to survive financially has become critical in maintaining and strengthening their financial resilience. By seeking sustainable financial goals, enterprises can also achieve sustainability.

In addition, in the face of different crises, companies can use the four dimensions of the BSCs to help companies focus more smoothly on the strategic goals they want to achieve. Before the crisis breaks out, being prepared for the crisis can most effectively reduce the losses caused by the crisis, and maintaining financial discipline is a good strategy for the enterprise. Maintaining a crisis preparedness fund and not investing blindly can help companies have a good financial backing in the event of a crisis. When the crisis breaks out, companies can choose to engage in more diversified business models, and seize this opportunity to record and analyze the value of each business well, because after the crisis, companies still have to pursue profits, find target customers, eliminate waste, and retain the business that allows the company to grow.

Although this study succeeded in constructing a practical crisis management model, there are still some limitations. The crisis management evaluation framework proposed in this paper is based on general guidelines, however, companies should take into account the characteristics of the company’s industry, industry profiles, and the differences of country conditions. They should consider the characteristics of enterprises and choose the most suitable crisis management tool. It would be worth exploring some interesting ideas in future work in order to extend this paper’s current analysis. For example, multiple comparisons with other data selection models can show how the model in this study is superior to other approaches. Moreover, doing so can help evaluate the appropriateness of enterprise crisis management under harsh times in the market as discussed in this study.

3.I suggest the authors emphasize how the architecture developed in this paper can be applied in other types of crises not only the ones generated by Covid pandemic

Response: Thanks for your valuable suggestions, we have made considerable modifications in this context. (Please see 6. Conclusion)

With the end of COVID-19 still far in sight and in the face of major and rapid eco-nomic changes, the strategy for enterprises to survive financially has become critical in maintaining and strengthening their financial resilience. By seeking sustainable finan-cial goals, enterprises can also achieve sustainability.

In addition, in the face of different crises, companies can use the four dimensions of the BSCs to help companies focus more smoothly on the strategic goals they want to achieve. Before the crisis breaks out, being prepared for the crisis can most effectively reduce the losses caused by the crisis, and maintaining financial discipline is a good strategy for the enterprise. Maintaining a crisis preparedness fund and not investing blindly can help companies have a good financial backing in the event of a crisis. When the crisis breaks out, companies can choose to engage in more diversified business models, and seize this opportunity to record and analyze the value of each business well, because after the crisis, companies still have to pursue profits, find target customers, eliminate waste, and retain the business that allows the company to grow.
